# Trademark Similarity Evaluation Using a Combination of ViT and Local Features

**Dmitry Vesnin** *,† , **Dmitry Levshun** *,† and **Andrey Chechulin** *,†

St. Petersburg Federal Research Center of the Russian Academy of Sciences (SPC RAS),
199178 St. Petersburg, Russia
* Correspondence: vesnin@comsec.spb.ru (D.V.); levshun@comsec.spb.ru (D.L.); chechulin@comsec.spb.ru (A.C.)
† These authors contributed equally to this work.

**Abstract:** The origin of the trademark similarity analysis problem lies within the legal area, specifically the protection of intellectual property. One of the possible technical solutions for this issue is the trademark similarity evaluation pipeline based on the content-based image retrieval approach. CNN-based off-the-shelf features have shown themselves as a good baseline for trademark retrieval. However, in recent years, the computer vision area has been transitioning from CNNs to a new architecture, namely, Vision Transformer. In this paper, we investigate the performance of off-the-shelf features extracted with vision transformers and explore the effects of pre-, post-processing, and pre-training on big datasets. We propose the enhancement of the trademark similarity evaluation pipeline by joint usage of global and local features, which leverages the best aspects of both approaches. Experimental results on the METU Trademark Dataset show that off-the-shelf features extracted with ViT-based models outperform off-the-shelf features from CNN-based models. The proposed method achieves a mAP value of 31.23, surpassing previous state-of-the-art results. We assume that the usage of an enhanced trademark similarity evaluation pipeline allows for the improvement of the protection of intellectual property with the help of artificial intelligence methods. Moreover, this approach enables one to identify cases of unfair use of such data and form an evidence base for litigation.

**Keywords:** trademarks; data protection; artificial intelligence; image processing; trademark retrieval

## 1. Introduction

A trademark (logo) is a company's most valuable intellectual property. Trademarks require registration to avoid reputational damage and damage to profits caused by trademark infringement. A trademark can only be registered if it is unique and unlike other registered trademarks.

An estimated 13.4 million trademark applications were filed worldwide in 2020. That is nearly 1.9 million more than were filed in 2019, an increase of 16.5% over the previous year. This high growth rate was achieved despite the onset of the COVID-19 pandemic and the following global economic downturn. This is also the eleventh consecutive year of growth since the end of the global financial crisis and a return to double-digit growth rates, up from 5.7% in 2019; see Figure 1 [1].

A quick way to search for similar logos is needed to prevent intellectual property theft. A manual search is practically impossible due to the number of registered trademarks and issues with scaling the search. This necessitates the development of methods to automate such searches. With the help of such methods, it is possible to create a system to help an expert search for similar logos.

The standard approach for finding similar logos is using off-the-shelf neural network features based on CNN architecture. In this paper, we compare the quality of off-the-shelf features extracted by convolutional neural network (CNN) and vision transformer (ViT), consider the effect of different pre- and post-processing techniques, and add local features to improve the final result. To the best of our knowledge, this is the first study that examines

the use of ViT for similar logo search tasks and the effect of pre- and post-processing on the quality of similar logo searches.

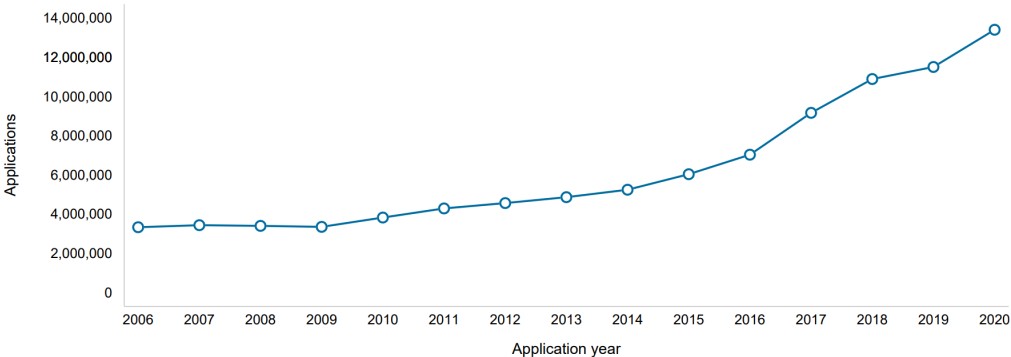

**Figure 1.** Chart of the number of applications for registration of new trademarks.

The origin of the trademark similarity analysis problem lies within the legal area, specifically the protection of intellectual property. Trademark infringement is a significant concern, as unauthorized use of a trademark can lead to reputational damage and financial losses for businesses. The legal aspect necessitates a technological solution to effectively detect and address trademark similarities. As a possible solution, the trademarks' similarity evaluation pipeline based on the content-based image retrieval (CBIR) approach is used to solve this legal issue. However, CBIR in such a context also poses some challenges. These challenges include a large search space, partial/semantic similarity, and limited computing resources. However, the task of trademark retrieval also introduces unique obstacles. Trademarks, being heavily stylized, contain less information compared to natural images and lack the rich texture commonly found in natural image content. Additionally, trademarks often share common design elements, such as characters and icons. Another complexity lies in the ambiguous and broad definition of trademark similarity (particularly in the legal area), which encompasses multiple aspects such as shape, layout, texture, and partial aspects. This paper addresses these issues by presenting possible technical solutions to enhance the pre-processing and post-processing steps of the trademarks' similarity evaluation pipeline.

This paper makes several contributions to the field of trademark retrieval. Firstly, it evaluates the performance of off-the-shelf features extracted with ViT compared to traditional CNN-based models, demonstrating the superiority of ViT-based models in trademark retrieval. Additionally, the paper proposes the joint utilizing global and local features, effectively combining their strengths to improve the overall search quality. Furthermore, the study investigates all steps of the trademark retrieval pipeline, including the effects of pre-, post-processing, and using models pre-trained on large datasets. Overall, the paper presents a comprehensive analysis and achieves state-of-the-art results in trademark retrieval.

This article consists of six sections. Here, in Section 1, we justified the need to search for similar logos and outlined the novelty of the work. In Section 2, we review the relevant literature and research directions. Section 3 presents an approach to image pre-processing for analyzing trademark similarity using ViT models and local features. In Section 4, we describe the dataset and experimental results and analyze the results. In Section 5, we discuss the search results using the developed method, and in Section 6, we present the conclusions and describe the directions for further research.

## 2. Related Work

The trademark retrieval task is a subtask of image retrieval that researchers have been working on for quite some time. The most common type of retrieval is content-based image retrieval (CBIR). This approach uses computer vision techniques to solve the challenge of finding similar images. The following features are used to search similar images: shape,

color, texture, local features (based on key points), global features (based on the use of neural networks), and others.

In [2], a dataset for testing the similarity search algorithms, namely, the METU Trademark Dataset, and a comparison of classical features (texture, shape, color histogram) and features generated by a neural network on the task of searching for similar trademarks are presented. The article shows that the classical features are significantly inferior to the features generated by the neural network. We considered AlexNet [3], GoogLeNet [4], and VGG16 [5], trained on the ImageNet dataset [6]. The fusion of all the models and methods considered in this article achieved a NAR of 0.062 ± 0.095 (Normalized average rank).

In [7], the authors fine-tuned two VGG19 models pre-trained on the ImageNet dataset. VGG19v was fine-tuned on data where images are visually similar, and VGG19c on conceptually similar images. A classification loss function was used, and the final NAR value was 0.047 ± 0.095.

The development of the trademark retrieval field goes hand in hand with the development of the image retrieval field. This is not surprising, since the search for similar trademarks is a subtask of the search for similar images.

In the following papers, the authors started experiments with Pooling layers: MAC, SPoC [8], CroW [9], R-MAC [10], and GeM [11]. These articles describe how to obtain better features from CNN layers using different Pooling layers.

VGG16 and MAC/SPoC/CRoW/R-MAC on the trademark search problem are discussed in [12]. It also deals with the issue that the neural network is "distracted" by the text in the trademark, which reduces the search quality. The authors developed two attention management methods to give more weight to the geometric component of the trademark, not the text. The best model uses R-MAC, and text removal has a mAP (mean Average Precision) value equal to 25.7.

In [13], the authors considered the possibility of applying the attention mechanism, CNN architecture, and unsupervised learning. The final model used ECANet50 and was trained with instance discrimination. The model showed a NAR of 0.051 ± 0.002.

In [14], the authors applied reinforcement learning to train an ensemble of TTA (test-time augmentation) image augmentation policies. This technique makes off-the-shelf features of CNN models more invariant to various image transformations. In this paper, the mAP metric equal to 30.5 was used.

In [15], improvements of R-MAC such as MR (Multi-Resolution), SMAC (sum and max activation of convolution), and URA (unsupervised regional attention) were considered. This work shows state-of-the-art results with a mAP of 31.0% and NAR of 0.028.

As can be seen from the articles listed above, only CNN-generated features and ways to improve their quality are mainly investigated. Local features are practically not used. At the same time, a promising architecture—ViT [16]—has not yet been fully tested. Since the ViT architecture has already been applied to image retrieval issues, we believe it is also reasonable to test its performance in trademark retrieval. It will also be useful to study the influence of different pre- and post-processing techniques on the features extracted with this architecture and their joint work with local features.

## 3. Pipeline Enhancement

In this section, we present improvements for the pipeline for solving the problem of searching for similar logos. The essence of changes is to apply local and global features to improve the search quality. A schematic of the pipeline is shown in Figure 2. Each step is numbered, and the elements inside the blocks represent the possible implementation of the steps. Next, we describe in detail the goals of each stage and the possible options for their implementation.

*Step 1: Pre-processing.* Pre-processing, and in particular the scaling of the image to a certain size, is an important step in the search for similar images because it is at this stage where we set the constraints with which our system will have to work. Image scaling will inevitably lead to loss of information, but we can minimize the negative effect.

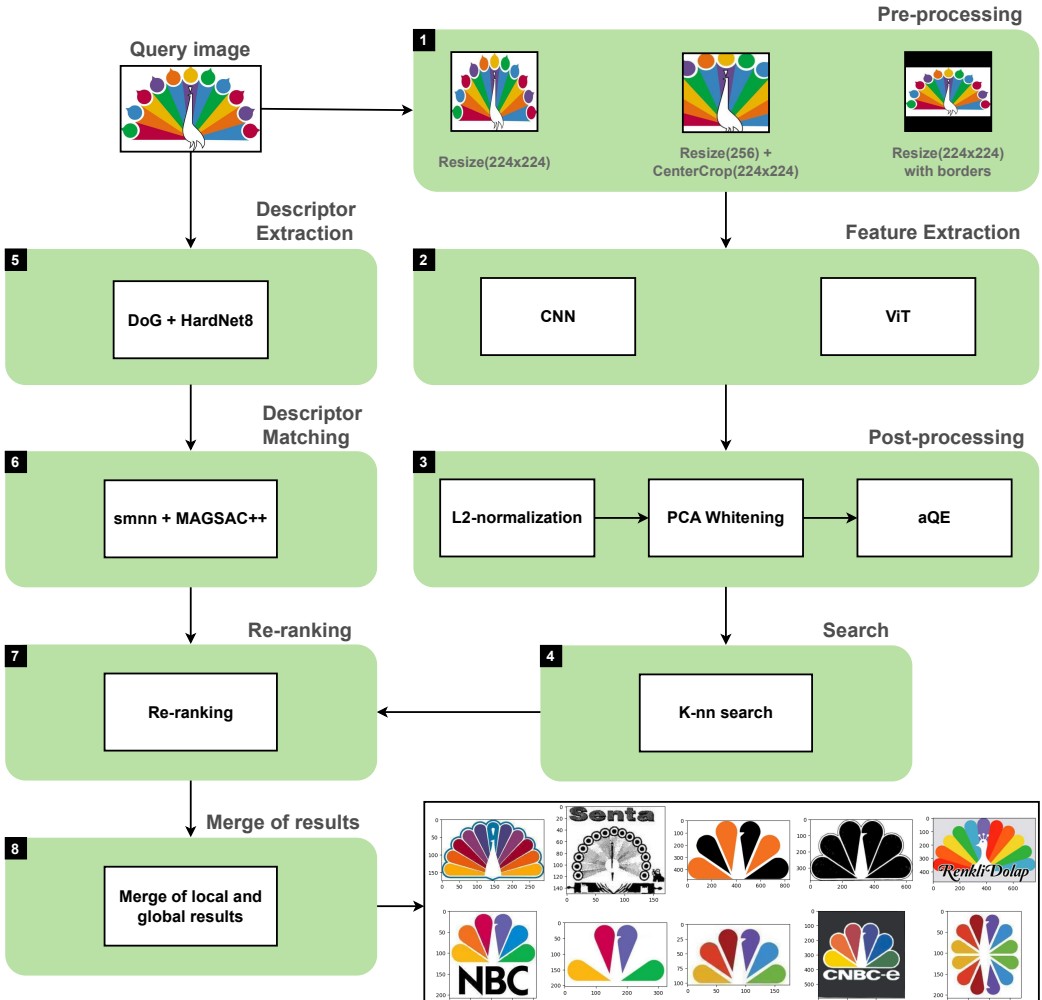

**Figure 2.** Pipeline overview.

Image scaling is needed because most neural networks are trained for image classification and work effectively only on images with a certain size—the size of images used in the training phase. The most commonly used image size is 224 × 224. There are several ways to scale the image to this size, see Figure 3:

- Scaling to 224 × 224 without preserving the aspect ratio (Resize (224 × 224)).
- Scaling to 256 with the aspect ratio preserved, then cropping out a 224 × 224 square from the center of the image (Resize (256) + CenterCrop (224 × 224)).
- Scaling to 224 × 224 with the aspect ratio preserved, where black bars appear (Resize (224 × 224) with borders).

Each method has its disadvantages:

- Resize (256) + CenterCrop (224 × 224) is a standard transformation used in classification. Its disadvantage is that part of the image is cropped, and thus information is lost.
- Resize (224 × 224) loses aspect ratio information, which makes the image look very different visually.
- Resize (224 × 224) with borders preserves the aspect ratio information but reduces the effective resolution.

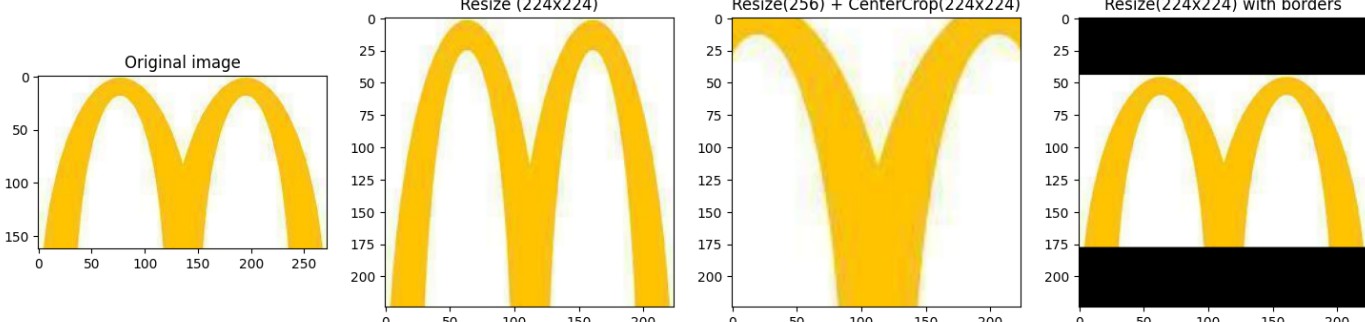

**Figure 3.** Examples of image scaling.

After scaling is applied, the images are normalized using the mean and standard deviation of the dataset on which the training occurred. The images are then fed to the feature extraction stage.

*Step 2. Feature extraction.* Global features are features that describe the entire image. Global features can summarize the entire image with a single vector describing color, texture, or shape. Examples of classical global features are GIST, LBP, and others. In [17], it was first observed that features extracted by CNNs trained on image classification datasets are good candidates for most visual recognition tasks, including finding similar images.

In this paper, we extract features using two architectures: CNN and ViT. The features were extracted for CNN-based networks using Global Max Pooling from the last Conv layer. For ViT, a CLS token was used. After feature extraction, post-processing takes place.

*Step 3. Post-processing.* Various techniques can be used to post-process the features obtained with neural networks to improve the quality of search results. In contrast to pre-processing, the techniques described below can be applied simultaneously in order of numbering. The proposed approach uses L2-normalization, PCA whitening, and $\alpha$QE. Let us take a closer look at them.

L2-normalization is a common technique for normalizing feature vectors. This technique divides each feature vector element by its L2-norm, which results in a unit vector of length 1. It reduces the effect of the feature vector's magnitude and improves the features' stability against changes in the image.

Query expansion (QE) is a process of reformulating a search query to improve search accuracy. Its essence is to use the information obtained from the primary search to modify the vector by which the search is conducted.

For the task of image retrieval, there are several methods of QE, such as AQE (Average Query Expansion) [18], $\alpha$QE ($\alpha$-weighted query expansion) [11], and others. AQE is based on averaging the features of the top-ranked images in the original search and using the averaged feature for a new query, while $\alpha$QE uses weighted averaging. Thus, when $\alpha = 0$, $\alpha$QE becomes equivalent to AQE.

*Step 4. Search.* After all post-processing stages, the resulting features are used to search for similar images. In the previous studies, a direct relationship between the similarity of vector representations of images and their visual similarity was found. Moreover, the closer the numerical values of the vectors of two images are to each other, the more likely these images will have similar visual characteristics. Therefore, in this paper, we use K-NN (K nearest neighbors) search to obtain a list of k best-fitting candidates. To measure the distance between vectors, we apply the distance metric L2, which allows us to determine the Euclidean distance between two vectors in n-dimensional space.

*Step 5. Descriptor Extraction.* Local features describe local parts of the image: neighborhoods of keypoints. Key points (keypoints) are special points or areas in the image with unique visual characteristics and relative invariance to changes in scale, orientation, illumination, and other affine transformations. A detector is used to find key points, and a descriptor is used to describe them. By comparing the keypoint descriptors, we can find out how similar the local parts of different images are. This step uses a combination of

the DoG detector and the HardNet8 descriptor, which has been shown to be successful in a number of works, e.g., in [19].

*Step 6. Descriptor Matching.* Descriptor matching is the process of finding the closest keypoint descriptors. SMNN (second mutual nearest neighbors) is used for descriptor matching. We also use RANSAC to filter outliers. In this paper, we use MAGSAC++ [20]. After this step, we obtain a list of candidates sorted by the number of inliers. In step 8, we will describe the merge of search results obtained using local and global features.

*Step 7. Re-ranking.* The results obtained with global features are re-ranked with local features. We use the same methods to extract and compare descriptors as in steps 5 and 6 and obtain the number of inliers. Then we calculate the re-ranking score using the following formula:

$$score_{reranking} = \frac{1}{distance + e} \times W_{global} + inliers \times W_{local}$$

where *distance* is L2 distance, *e* is a small number to prevent division by zero, *inliers* is a number of inliers, and $W_{local}$ and $W_{global}$ are weights. The output is a ranked list of candidates, which is combined with local search results in the next step.

*Step 8. Merging of results.* The main disadvantage of the global feature approach is its inability to pay attention to smaller parts of the image. For example, even if an element is exactly the same as another, if is not the main element in the image, it will be closer to the bottom of the ranked list. To overcome this issue, we combine local and global search results. To combine the two search results, we use the following equation to calculate the final score:

$$score_{joint} = score_{reranking} \times W_{joint} + inliers \times W_{local}$$

If the candidate is not on one of the lists, the score for this parameter is zero. In the next section, we will consider the effect of the steps in our pipeline on the quality of the search for similar logos.

## 4. Experimental Evaluation

In this section, we consider the effect of pre-processing and post-processing on the features generated by the CNN and ViT architecture models, as well as the results of using local features for re-ranking and joint usage of global and local features.

All experiments were conducted on a PC with Intel Core i5-9600 processor, 32 GB DDR4 RAM, and NVIDIA GeForce RTX 2060 SUPER graphics card.

Tables 1 and 2 show the tested models of two architectures: CNN and ViT. They also indicate on which dataset pre-training was performed. CNN models are represented by ResNet [21]. ViT models are represented by ViT [16] and BEiT [22].

**Table 1.** CNN Models.

| Model Name | Pre-Training Dataset |
|---|---|
| Resnet18_in1k | ResNet18 pre-trained on ImageNet-1K |
| Resnet50_in1k | ResNet50 pre-trained on ImageNet-1K |
| resnetv2_50x1_bitm_in21k | ResNet50V2 pre-trained on ImageNet-21K |

**Table 2.** ViT Models.

| Model Name | Pre-Training Dataset |
|---|---|
| ViT B/16 in 1k | ViT B/16 pre-trained on ImageNet-1K |
| ViT B/16 in 21k | ViT B/16 pre-trained on ImageNet-21K |
| BEiT ViT B/16 in 21k | BEiT ViT B/16 pre-trained on ImageNet-21K |

The tests were performed on the METU Trademark Dataset [17], which is the largest dataset for the trademark retrieval task. It has two main sets: the query set and the test set. The test set is unlabeled and contains more than 900,000 images. The query set contains 417 trademark images that belong to 35 different classes; some examples are shown in Figures 4 and 5.

For the evaluation, we injected the query set (labeled) into the test set (unlabeled). Unlabeled data act as a distraction to make it harder to find relevant trademark logos among the query set. The same protocol is used in other works, for example, [2].

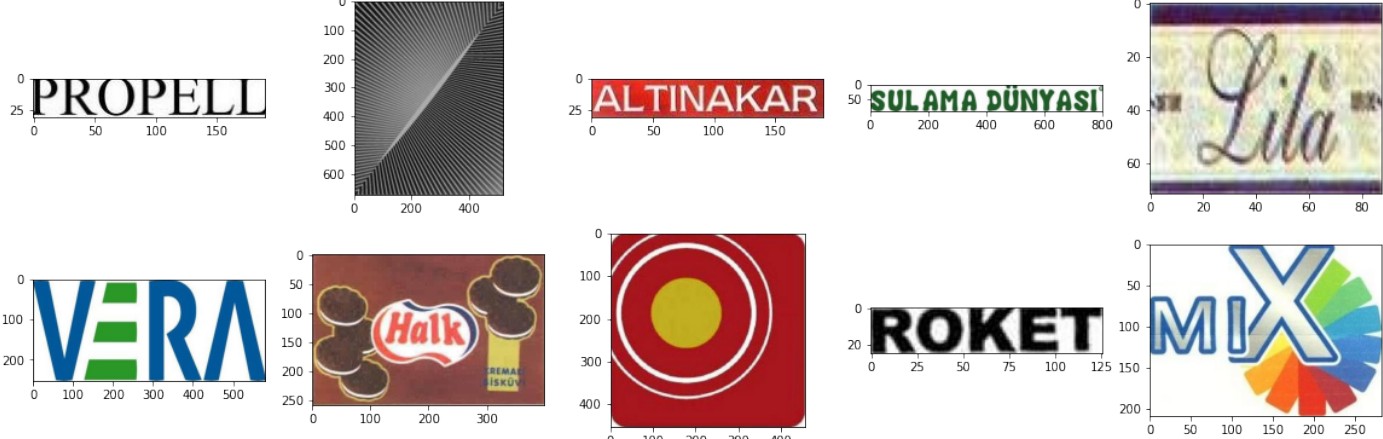

**Figure 4.** Example of dataset samples.

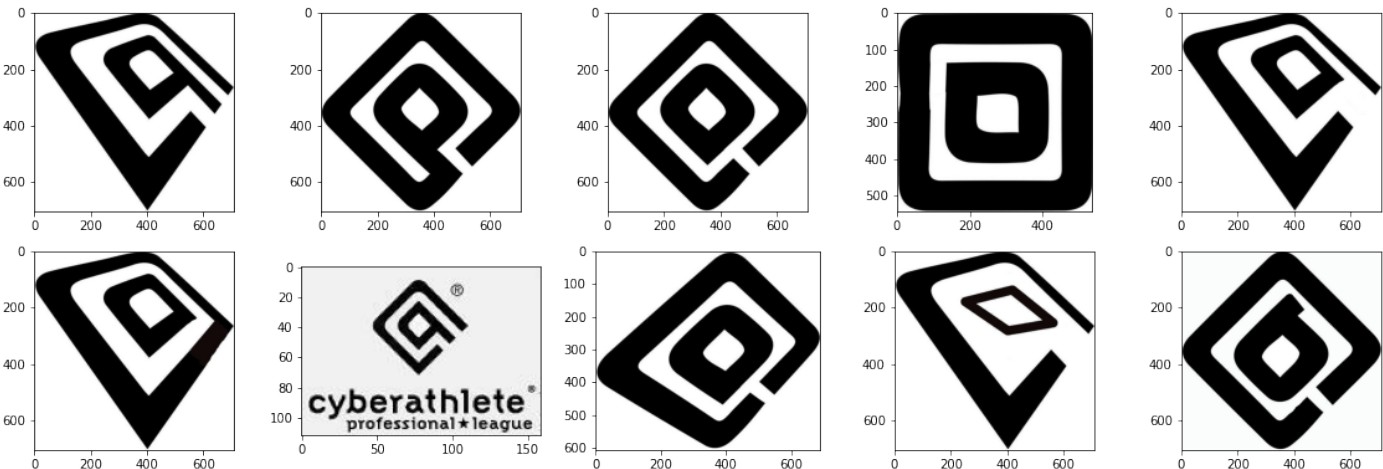

**Figure 5.** Example of similar trademarks from query set.

For the evaluation, we used the mAP value:

$$mAP@k = \frac{1}{N}\sum_{i}^{N} AP@k_i$$

It is calculated by using the top *k* retrieved results, where *k* is 100, and *AP@k* is the average precision at *k*. Table 3 and Figure 6 show the mean average precision of CNN and ViT-based models using various resizing techniques without post-processing steps.

From the results of the experiments, we can draw several conclusions. First of all, versions of neural networks pre-trained on larger datasets show greater mAP.

All models, except resnet18_in_1k and ViT B/16 in 1k, have the highest mAP when using Resize (224 × 224) with borders. This shows that the aspect ratio information lost with other types of Resize can be used by neural networks for more accurate search results.

ViT architecture loses to CNN architecture when pre-trained on small datasets (ImageNet 1K) but wins when pre-trained on larger datasets (ImageNet 21K). This shows that, with increasing dataset size, ViT features become more useful in solving the problem of searching for similar logos.

**Table 3.** Mean average precision of CNN- and ViT-based models.

| Neural Network | Dimensions | Resizing Technique | mAP@100 |
|---|---|---|---|
| Resnet18_in1k | 512 | Resize (256) + CenterCrop (224 × 224) | 11.05 |
| | | Resize (224 × 224) | 12.23 |
| | | Resize (224 × 224) with borders | 11.25 |
| Resnet50_in1k | 2048 | Resize (256) + CenterCrop (224 × 224) | 11.55 |
| | | Resize (224 × 224) | 13.19 |
| | | Resize (224 × 224) with borders | 13.62 |
| resnetv2_50x1_bitm_in21k | 2048 | Resize (256) + CenterCrop (224 × 224) | 14.02 |
| | | Resize (224 × 224) | 14.93 |
| | | Resize (224 × 224) with borders | 15.17 |
| ViT B/16 in 1k | 768 | Resize (256) + CenterCrop (224 × 224) | 7.44 |
| | | Resize (224 × 224) | 8.30 |
| | | Resize (224 × 224) with borders | 7.35 |
| ViT B/16 in 21k | 768 | Resize (256) + CenterCrop (224 × 224) | 14.65 |
| | | Resize (224 × 224) | 15.97 |
| | | Resize (224 × 224) with borders | 16.83 |
| BEiT ViT B/16 in 21k | 768 | Resize (256) + CenterCrop (224 × 224) | 18.01 |
| | | Resize (224 × 224) | 19.63 |
| | | Resize (224 × 224) with borders | 20.19 |

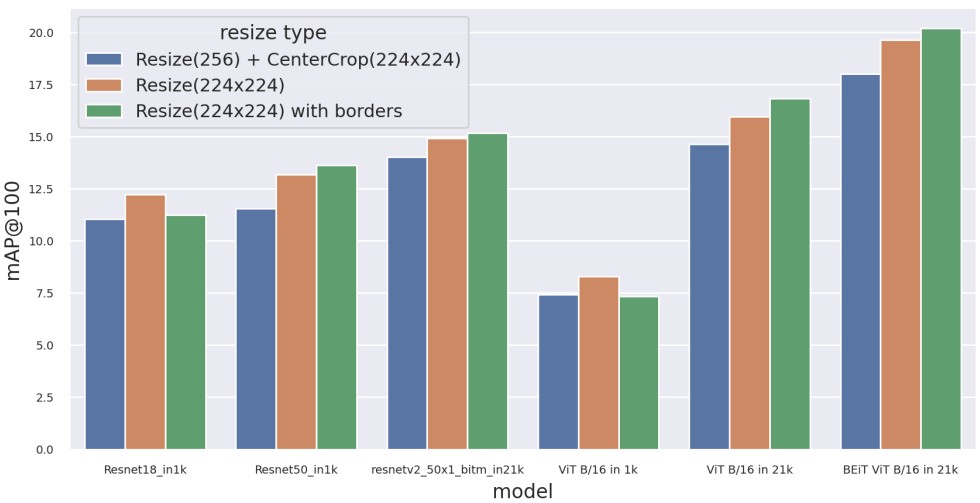

**Figure 6.** Mean average precision of CNN and ViT-based models.

Table 4 shows the results of applying PCAw to resnetv2_50x1_bitm_in21k and BEiT ViT B/16 in 21k, which are, respectively, the best CNN-based and ViT-based models. The resizing technique used was Resize (224 × 224) with borders. We can see that PCAw positively impacted the mAP value and benefited the ViT-based model more than the CNN-based model.

**Table 4.** Results of applying PCAw.

| Neural Network | mAP@100 before Applying PCAw | mAP@100 after Applying PCAw |
|---|---|---|
| resnetv2_50x1_bitm_in21k | 15.17 | 18.50 |
| BEiT ViT B/16 in 21k | 20.19 | 25.12 |

Figure 7 shows the impact of aQE on the performance of BEiT ViT B/16 in 21k + PCAw. The best result of 28.46 is achieved with alpha = 0.5 and n = 4, which is slightly better than AQE (alpha = 0).

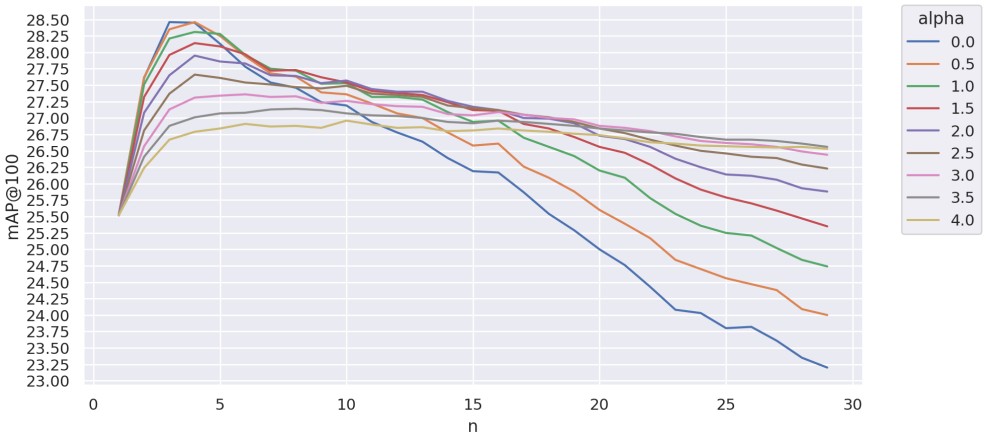

**Figure 7.** Impact of aQE on the mAP value.

Table 5 shows the cumulative effect of post-processing and local features on the mAP value. The usage of pcaW gave the greatest increase in search accuracy.

**Table 5.** Cumulative effect of post-processing and local features on the mAP value.

| Method | mAP@100 |
|---|---|
| BEiT ViT B/16 in 21k | 20.19 |
| BEiT ViT B/16 in 21k + pcaW | 25.12 |
| BEiT ViT B/16 in 21k + pcaW + aQE | 28.46 |
| BEiT ViT B/16 in 21k + pcaW + aQE + reranking | 30.62 |
| BEiT ViT B/16 in 21k + pcaW + aQE + reranking + local_features | 31.23 |

Table 6 presents a comparison of our approach with other works. Our approach is slightly ahead of the current state-of-the-art result shown by MR-R-SMAC w/URA.

**Table 6.** Comparison of our approach with other works.

| Method | mAP@100 |
|---|---|
| SPoC [12] | 18.7 |
| CAM MAC [12] | 22.3 |
| ATRHA R-MAC [12] | 25.7 |
| TTA [14] | 30.5 |
| MR-R-SMAC w/URA [16] | 31.0 |
| BEiT ViT B/16 in 21k + pcaW + aQE + reranking + local_features | 31.23 |

The experimental results show that ViT networks pre-trained on large datasets can show better results than CNNs, and they do not require specialized Pooling layers. We can

significantly increase search accuracy by using efficient pre- and post-processing. Using local features improves search results, suggesting that off-the-shelf features extracted by neural networks can still be improved.

## 5. Discussion

Despite the combined use of global and local features and advanced architectures like ViT and modern descriptors like HardNet8, the results still need to be better.

As a result of the tests, we have identified some issues that our method faces. Figure 8 shows examples of searches; here, yellow frames show images that similar to the searched image but are absent from the test dataset, and red frames show false positives.

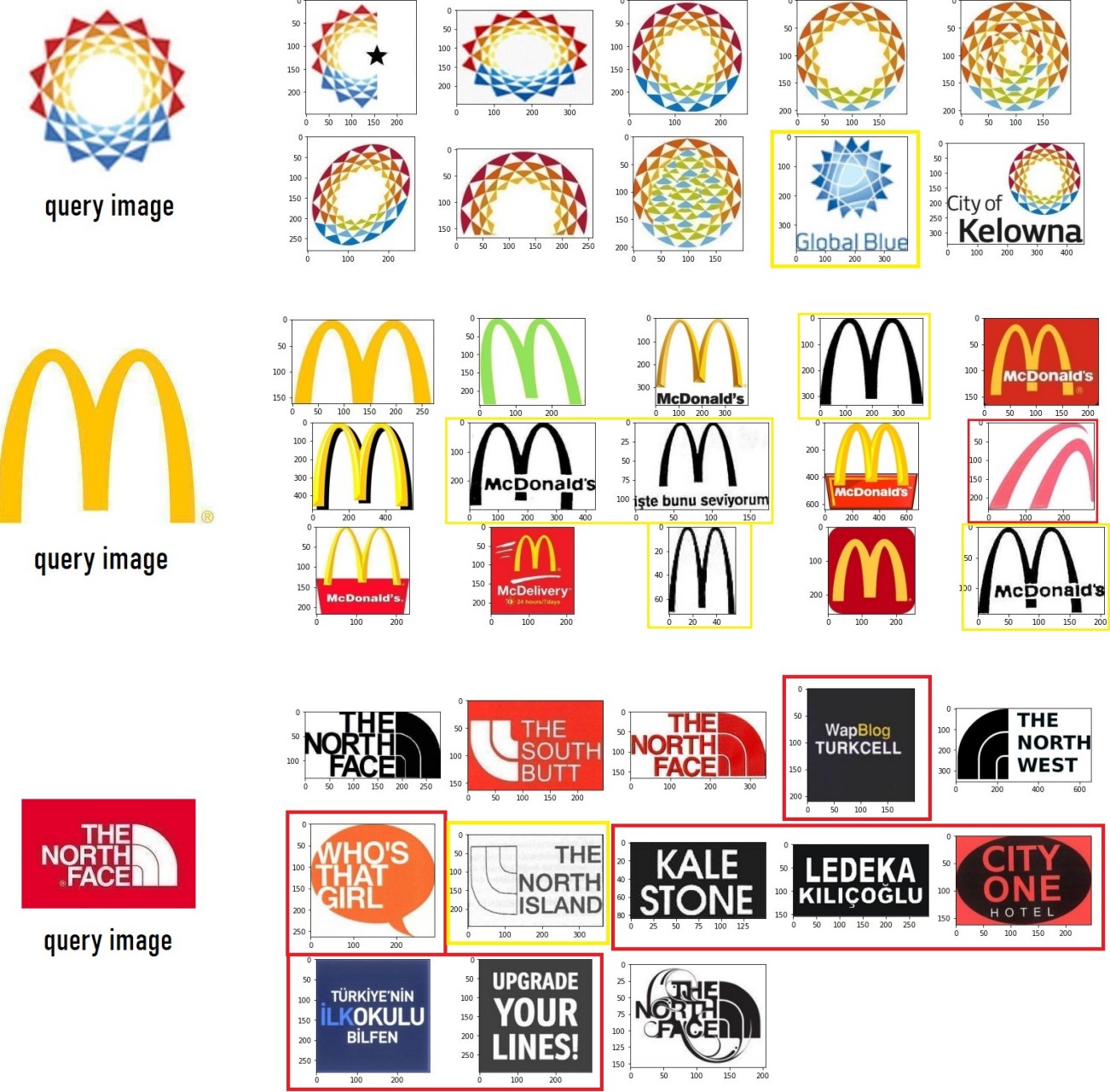

**Figure 8.** Examples of searches.

While analyzing the results, we noticed that the system can find images that seem similar but are not in the query set of the METU Trademark dataset. The test set should be extended. The presence of text on a logo significantly degrades search results for

several reasons. With the global features, we are not yet able to distinguish the "important" and "unimportant" parts of the image because, as shown in Figure 8, the search results contain pictures with a similar layout or similar font but do not take into account a similar geometric element of the logo. Also interesting is that, in some cases, the difference in distance between similar and dissimilar images is minimal, indicating that it is quite difficult to distinguish between similar and dissimilar images with the current features.

A similar issue is observed with local features. The text in the image receives a lot of keypoints, due to which there may be cases of false positives, as the letters are similar to each other, i.e., local features find images with similar text font.

These issues can be solved as follows: it is necessary to perform fine-tuning of the neural network, with which the global features are extracted, and limit the number of keypoints in the area around the text in the image.

## 6. Conclusions

In this paper, we have presented an approach for searching for similar logos and performed exhaustive testing of all its steps. We found that, for similar logos searching, the best type of image scaling is scaling with aspect ratio preservation. Despite the decreased effective resolution, neural networks benefit from retaining information about the original image such as proportions. For the BEiT model, the gain was 2.1% mAP. A comparison of the features extracted from ViT- and CNN-based models was performed, and many interesting properties were found: ViT-based models are inferior to CNN-based models when pre-trained on the smaller dataset (imagenet-1k) but start to win with the increasing dataset. The difference between the best CNN and ViT models, without post-processing, is 5% mAP. It was also concluded that PCA whitening works not only on features derived from CNN models but also on models based on Vision Transformers, with ViT having a higher mAP gain than CNN (3.3% for ResnetV2 and 4.9% for BEiT). When optimal parameters are chosen, aQE can perform better than AQE. A further 3.3% gain was achieved with aQE. This paper demonstrates the use of joint global and local features. Local features have been used twice: for re-ranking results obtained with global features and for merging local and global results. This has further improved the search quality, reaching a mAP value of 31.23%.

In future work, we plan to study metric learning for training models based on the Vision Transformer and deeper integration of local and global features. Moreover, it is planned to implement this approach as a digital forensics tool as a part of data analysis services of the International Digital Forensics Center [23].

**Author Contributions:** Conceptualization, D.V., D.L. and A.C.; methodology, D.V., D.L. and A.C.; software, D.V.; validation, D.V., D.L. and A.C.; formal analysis, D.L. and A.C.; investigation, D.V. and A.C.; resources, D.V.; data curation, A.C.; writing—original draft preparation, D.V.; writing—review and editing, D.L. and A.C.; visualization, D.V. and D.L.; supervision, A.C.; project administration, A.C.; funding acquisition, A.C. All authors have read and agreed to the published version of the manuscript.

**Funding:** The reported study was partially funded by the budget project FFZF-2022-0007.

**Institutional Review Board Statement:** Not applicable.

**Informed Consent Statement:** Not applicable.

**Data Availability Statement:** No new data were created or analyzed in this study. Data sharing is not applicable to this article.

**Conflicts of Interest:** The authors declare no conflict of interest.

## Abbreviations

The following abbreviations are used in this manuscript:

| | |
|---|---|
| ViT | Vision transformer |
| CNN | Convolutional neural network |
| CBIR | Content-based image retrieval |
| mAP | Mean average precision |
| NAR | Normalized average rank |
| MR | Multi-resolution |
| SMAC | Sum and max activation of convolution |
| URA | Unsupervised regional attention |
| AQE | Average query expansion |
| $\alpha$QE | $\alpha$-weighted query expansion |
| K-NN | K nearest neighbors |
| SMNN | Second mutual nearest neighbors |

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
