# Peer review of "Trademark Similarity Evaluation Using a Combination of ViT and Local Features"

_information, doi:10.3390/info14070398_

Round 1

Reviewer 1 Report

The authors developed techniques based on CNN and ViT and their combination to detect similar trademark logos.  The approach seems correct and technically sound.  I only ask the authors to justify their work and explain how the application (trademark logo similarity) is different from matching the similarity of any general image to a database of images. What is it about the trademark logos that would require special treatment?

Author Response

Dear reviewer,

Thanks a lot for your time and effort to evaluate our manuscript. During the analysis of the feedback, we used your comments to improve our work. Detailed responses are presented below.

Comment. I only ask the authors to justify their work and explain how the application (trademark logo similarity) is different from matching the similarity of any general image to a database of images. What is it about the trademark logos that would require special treatment?

Response. The task of defining trademark logo similarity is differing from general image matching, because logo similarity currently is defined mostly by jurists and the problem lies within the legal area. For example, having common design elements does not necessarily mean that logos are similar. Trademark similarity can be based on a lot of factors including shape, geometry, texture, partial elements and others. Moreover, for the trademark retrieval task, currently there are no open datasets labeled by experts, with the help of which we could fine-tune models. Therefore, we have to use pre-trained models and various methods for pre- and post-processing. 

To address this comment, we added the following text to the Introduction section:

The origin of the trademark similarity analysis problem lies within the legal area, specifically the protection of intellectual property. Trademark infringement is a significant concern, as unauthorized use can lead to reputational damage and financial losses for businesses. The legal aspect necessitates a technological solution to effectively detect and address trademark similarities. As a possible solution, the trademarks' similarity evaluation pipeline based on the content-based image retrieval (CBIR) approach is used to solve this legal issue. In turn, CBIR in such a context also poses some challenges. These challenges include a large search space, partial/semantic similarity, and limited computing resources. However, the task of trademark retrieval also introduces unique obstacles. Trademarks, being heavily stylized, contain less information compared to natural images and lack the rich texture commonly found in natural image content. Additionally, trademarks often share common design elements, such as characters and icons. Another complexity lies in the ambiguous and broad definition of trademark similarity (particularly in the legal area), which encompasses multiple aspects such as shape, layout, texture, and partial aspects. This paper addresses these issues by presenting possible technical solutions to enhance the pre-processing and post-processing steps of the trademarks' similarity evaluation pipeline.

Reviewer 2 Report

The title of the paper is attracting, but the content is too experimental, evaluating two NNs for similar logo searching.

So, the novelty of the paper as a methodology is very low. Also, there are evident meaning sentences such as:

". We assume that the proposed approach for the trademark similarity
evaluation will allow one to improve the protection of such data with the help of artificial intelligence methods"

Such sentences do not bring any valuable information.

Something very basic that never to do in ML:
"For the evaluation, the query set is injected into the test set."

"While analyzing the results, we noticed that the system can find images that seem similar but are not in the test set of the METU Trademark dataset."

"In future work, we plan to study metric learning for training models based on the Vision Transformer and deeper integration of local and global features. Also it is planned to implement this approach as a digital forensics tool as a part of data analysis services of the International Digital Forensics Center"

I think that it will better to wait and to include these materials and re-submit the paper.

Some improvement is needed.

Author Response

Dear reviewer,

Thanks a lot for your time and effort to evaluate our manuscript. During the analysis of the feedback, we used your comments to improve our work. Detailed responses are presented below.

Comment 1. The title of the paper is attracting, but the content is too experimental, evaluating two NNs for similar logo searching. So, the novelty of the paper as a methodology is very low.

Response. In our work, we investigate the impact of various types of pre- and post-processing on different architectures of neural networks on their final performance. To the best of our knowledge, it is the first work that examines in detail the impact of each of the stages of the trademark retrieval pipeline, their individual and combined impact on performance. We also show that features generated using ViT architecture are not worse, and sometimes even better, than features generated using CNN. We believe that the results of our work are useful for future research.

To address this comment, we added the following text to the Introduction section:

This paper makes several contributions to the field of trademark retrieval. Firstly, it evaluates the performance of off-the-shelf features extracted with ViT compared to traditional CNN-based models, demonstrating the superiority of ViT-based models in trademark retrieval. Additionally, the paper proposes the joint utilizing global and local features, effectively combining their strengths to improve the overall search quality. Furthermore, the study investigates all steps of the trademark retrieval pipeline, including the effects of pre, post-processing, and using models pre-trained on large datasets. Overall, the paper presents a comprehensive analysis and achieves state-of-the-art results in trademark retrieval.

Comment 2. There are evident meaning sentences such as: “We assume that the proposed approach for the trademark similarity evaluation will allow one to improve the protection of such data with the help of artificial intelligence methods” Such sentences do not bring any valuable information.

Response. To address this comment, we removed such sentences and updated the abstract of the manuscript: 

The origin of the trademark similarity analysis problem lies within the legal area, specifically the protection of intellectual property. One of the possible technical solutions for this issue is the trademark similarity evaluation pipeline based on the content-based image retrieval approach. CNN-based off-the-shelf features have shown themselves as a good baseline for trademark retrieval. However, in recent years, the computer vision area was transitioning from CNNs to a new architecture – Vision Transformer. In this paper, we investigate the performance of off-the-shelf features extracted with vision transformers and explore the effects of pre, post-processing, and pre-training on big datasets. We propose the enhancement of the trademark similarity evaluation pipeline by joint usage of global and local features, which leverages the best aspects of both approaches. Experimental results on METU Trademark Dataset show that off-the-shelf features extracted with ViT-based models outperform off-the-shelf features from CNN-based models. The proposed method achieves the mAP value of 31.23, surpassing previous state-of-the-art results. We assume that the usage of enhanced trademark similarity evaluation pipeline allows one to improve the protection of intellectual property with the help of artificial intelligence methods. Moreover, this approach allows one to identify cases of unfair use of such data and form an evidence base for litigation.

Comment 3. Something very basic that never to do in ML: “For the evaluation, the query set is injected into the test set.” “While analyzing the results, we noticed that the system can find images that seem similar but are not in the test set of the METU Trademark dataset.”

Response. The METU Trademark Dataset has two parts: query set and test set. The test set is unlabeled and used as a “distraction” for the query set. Query set consists of 35 query groups, each containing around 10 logos, which are similar to each other (similarity is defined by experts). We injected the query set into the test set before evaluation, so we can measure the mAP of our search results. Models do not observe the query set or the test set before the evaluation. The same protocol is used in other works, for example, [https://arxiv.org/pdf/1701.05766.pdf].

To make our decisions more clear, we added the following text to the Experimental evaluation section of the manuscript:

For the evaluation, we inject the query set (labeled) into the test set (unlabeled). Unlabeled data acts as a distraction to make it harder to find relevant trademark logos among the query set. The same protocol is used in other works, for example, [2]. 

Comment 4. “In future work, we plan to study metric learning for training models based on the Vision Transformer and deeper integration of local and global features. Also, it is planned to implement this approach as a digital forensics tool as a part of data analysis services of the International Digital Forensics Center” I think that it will better to wait and to include these materials and re-submit the paper.

Response. We believe that the current work can be published as a standalone paper. It investigates the performance of off-the-shelf features from Vision Transformers, proposes a method for combining global and local features, explores the impact of pre, post-processing, and pre-training on large datasets, and achieves state-of-the-art results. Moreover, currently there are no publicly available labeled dataset for fine-tuning, thus mentioned future work significantly differs from the focus and scope of the current paper.